# Strong electronic interaction and multiple quantum Hall ferromagnetic phases in trilayer graphene

Biswajit Datta[1], Santanu Dey[2], Abhisek Samanta[3], Hitesh Agarwal[1], Abhinandan Borah[1], Kenji Watanabe[4], Takashi Taniguchi[4], Rajdeep Sensarma[3] & Mandar M. Deshmukh[1]

Quantum Hall effect provides a simple way to study the competition between single particle physics and electronic interaction. However, electronic interaction becomes important only in very clean graphene samples and so far the trilayer graphene experiments are understood within non-interacting electron picture. Here, we report evidence of strong electronic interactions and quantum Hall ferromagnetism seen in Bernal-stacked trilayer graphene. Due to high mobility $\sim 500,000\,cm^2\,V^{-1}\,s^{-1}$ in our device compared to previous studies, we find all symmetry broken states and that Landau-level gaps are enhanced by interactions; an aspect explained by our self-consistent Hartree–Fock calculations. Moreover, we observe hysteresis as a function of filling factor and spikes in the longitudinal resistance which, together, signal the formation of quantum Hall ferromagnetic states at low magnetic field.

[1] Department of Condensed Matter Physics and Materials Science, Tata Institute of Fundamental Research, Homi Bhabha Road, Mumbai 400005, India. [2] Department of Astronomy and Astrophysics, Tata Institute of Fundamental Research, Homi Bhabha Road, Mumbai 400005, India. [3] Department of Theoretical Physics, Tata Institute of Fundamental Research, Homi Bhabha Road, Mumbai 400005, India. [4] Advanced Materials Laboratory, National Institute for Materials Science, 1-1 Namiki, Tsukuba 305-0044, Japan. Correspondence and requests for materials should be addressed to R.S. (email: sensarma@theory.tifr.res.in) or to M.M.D. (email: deshmukh@tifr.res.in).

Mesoscopic experiments tuning the relative importance of electronic interactions to observe complex ordered phases have a rich past[1]. While one class of experiments were conducted on bilayer two-dimensional electron systems (2DES) realized in semiconductor heterostructures, the other class of experiments focussed on probing multiple interacting sub-bands in quantum well structures[2]. There is an increasing interest in the electronic properties of few-layer graphene[3–13] as it offers a platform to study electronic interactions because the dispersion of bands can be tuned with number and stacking of layers in combination with electric field. Bernal/ABA-stacked trilayer graphene (ABA-TLG) provides a natural platform to observe such multi-subband physics as the band structure gives rise to monolayer-like (ML) and bilayer-like (BL) bands. The presence of the multiple bands and their Dirac nature lead to the possibility of observing an interesting interplay of electronic interactions in different channels leading to novel phases of the quantum Hall state.

Here we study the Landau-level (LL) spectrum on edge contacted ABA-TLG samples encapsulated in hexagonal boron nitride (hBN) flakes. We observe the coexistence of both massless and massive Dirac fermions in the form of parabolically dispersed LL crossing points at low magnetic field. At intermediate magnetic field we show that the LL fan diagram indicates that the electron-electron interactions lead to formation of symmetry broken spin and valley-polarized states. Our self-consistent Hartree–Fock calculation supports the observed interaction enhanced LL gaps at the symmetry broken states. We also observe hysteretic transport showing the formation of quantum Hall ferromagnetic (QHF) states.

## Results

**Magnetotransport in ABA-trilayer graphene.** Figure 1a shows the lattice structure of ABA-TLG with all the hopping parameters. We use Slonczewski–Weiss–McClure (SWMcC) parametrization of the tight binding model for ABA-TLG[14,15] (with hopping parameters $\gamma_0$, $\gamma_1$, $\gamma_2$, $\gamma_5$ and $\delta$) to calculate its low energy band structure. Definitions of all the hopping parameters are evident from Fig. 1a, and $\delta$ is the onsite energy difference of two inequivalent carbon atoms on the same layer. Its band structure, shown in Fig. 1b consists of both ML linear and BL quadratic bands[16,17].

Figure 1c shows an optical image of the device where the ABA-TLG graphene is encapsulated between two hBN flakes[18].

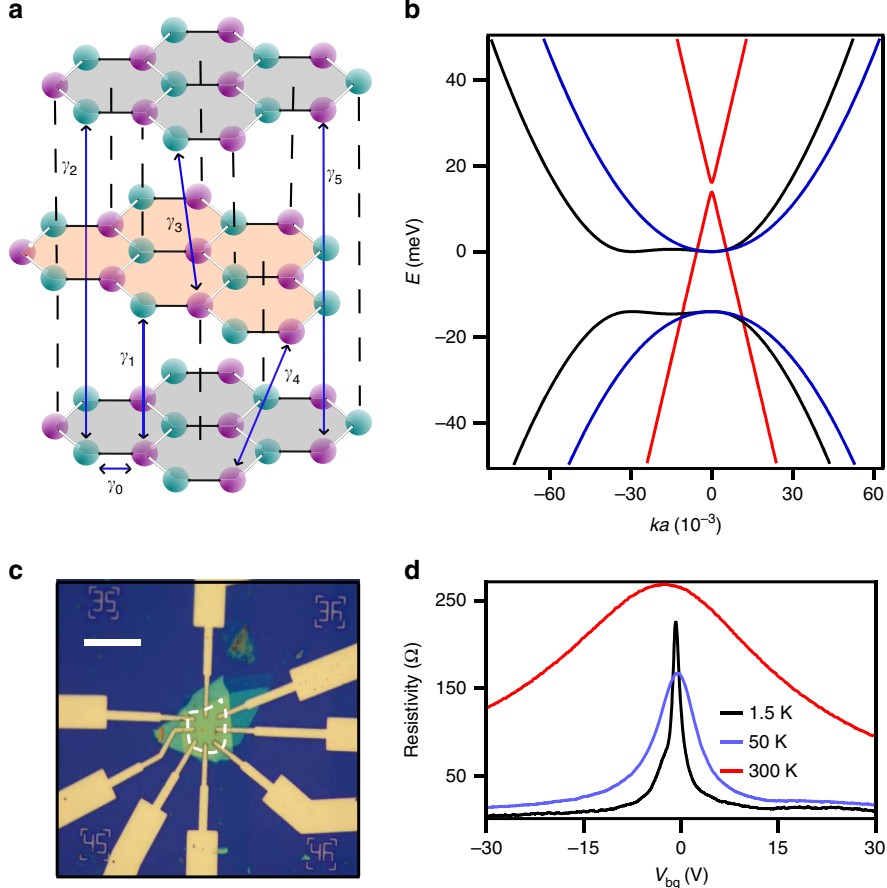

**Figure 1 | ABA-stacked trilayer graphene device and its electrical transport.** (**a**) Schematic of the crystal structure of ABA-TLG with all hopping parameters. (**b**) Low energy band structure of ABA-TLG around $k\_$ point ($-\frac{4\pi}{3}$, 0) in the Brillouin zone. The wave vector is normalized with the inverse of the lattice constant ($a = 2.46$ Å) of graphene. Black and blue lines denote the BL bands along $k_x$ and $k_y$ direction in the Brillouin zone whereas the red line denotes the ML band along both $k_x$ and $k_y$. ML bands are separated by $\sim \delta + \frac{\gamma_2}{2} - \frac{\gamma_5}{2} = 2$ meV and BL bands are separated by $\sim \frac{|\gamma_2|}{2} = 14$ meV. However, there is no band gap in total, semi-metallic nature of ABA-TLG is clear from the band overlap. (**c**) Optical image of the hBN encapsulated trilayer graphene device; Scale bar, 20 μm. White dashed line indicates the boundary of the ABA-TLG. The graphene sample is a slightly distorted rectangle, but the electrodes are designed in a Hall bar geometry. Length and breadth wise distance between furthest electrodes are 9.3 μm and 7.8 μm respectively. This makes the aspect ratio to be 1.19. Substrate consists of 30 nm thick hBN and 300 nm thick Silicon dioxide (SiO$_2$) coated highly $p$-doped silicon, which also serves as a global back gate. (**d**) Room temperature and low temperature four-probe resistivity of the device as a function of $V_{bg}$.

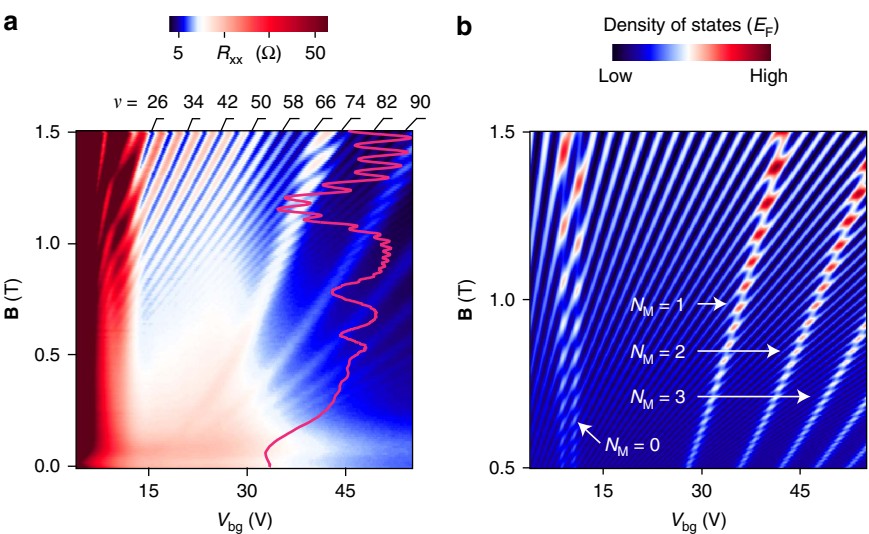

**Figure 2 | Low magnetic field fan diagram.** (**a**) Colour plot of $R_{xx}$ as a function of $V_{bg}$ and **B** up to 1.5 T. The LL crossings arising from ML bands and BL bands are clearly seen. Each parabola is formed by the repetitive crossings of a particular ML LL with other BL LLs. Crossing between any two LLs shows up as $R_{xx}$ maxima in transport measurement due to high DOS at the crossing points. The overlaid magenta line shows a line slice at $V_{bg} = 50$ V. (**b**) DOS corresponding to **a**. $N_M = 1$ labelled parabola refers to all the crossing points arising from the crossings of $N_M = 1$ LL with other BL LLs. Other labels have a similar meaning. $N_M = 0$ LL does not disperse with **B**, hence the crossings form a straight line parallel to **B** axis. Minimum **B** is taken as 0.5 T to keep finite number of LLs in the calculation. The horizontal axis is converted from charge density to an equivalent $V_{bg}$ after normalizing it by the capacitance per unit area ($C_{bg}$) for the ease of comparison with experimental fan diagram. $C_{bg}$ is determined from the high **B** quantum Hall data which matches well with the geometrical capacitance per unit area of 30 nm hBN and 300 nm SiO$_2$: $C_{bg} \sim 105 \, \mu Fm^{-2}$.

Four-probe resistivity ($\rho$) of the device is shown in Fig. 1d. The low disorder in the device is reflected in high mobility $\sim 500,000 \, cm^2 V^{-1} s^{-1}$ on electron side and $\sim 800,000 \, cm^2 V^{-1} s^{-1}$ on hole side; this leads to carrier mean free path in excess of $7 \, \mu m$ (see Supplementary Fig. 1 and Supplementary Note 1 for mobility and mean free path calculation). We measured one single gated device and one dual gated device on which we studied the effect of electric field. We found that the electron side data is relatively insensitive for low electric field range ($< 0.01 \, Vnm^{-1}$) (see Supplementary Fig. 2 and Supplementary Note 2 for dual gate device data). Due to the better quality of the single gated device we show the measurements done on the single gated device throughout the paper.

We next consider the magnetotransport in ABA-TLG that reveals the presence of LLs arising from both ML and BL bands. The LLs are characterized by the following quantum numbers: $N_M$ ($N_B$) defines the LL index with M (B) indicating monolayer (bilayer)-like LLs, $+ (-)$ denotes the valley index of the LLs and $\uparrow (\downarrow)$ denotes the spin quantum number of the electrons. All the data shown in this paper, are taken at 1.5 K. Figure 2a shows the measured longitudinal resistance ($R_{xx}$) as a function of gate voltage ($V_{bg}$) and magnetic field (**B**) in the low **B** regime (see Supplementary Fig. 3 and Supplementary Note 3 for more data at low magnetic field). Observation of LLs up to very high filling factor $\nu = 118$ confirms the high quality of the device. Along with the usual straight lines in the fan diagram, we find additional interesting parabolic lines which arise because of LL crossings. Figure 2b shows the calculated non-interacting density of states (DOS) in the same parameter range which matches very well with the measured resistance. We find that the low **B** data can be well understood in terms of non-interacting picture and it allows determination of the band parameters.

We now consider the LL fan diagram for a larger range of $V_{bg}$ and **B**. Figure 3a shows the calculated[14–16] energy dispersion of the spin degenerate LLs with **B**. All the band parameters of

multilayer graphene are not known precisely, so, we refine relatively smaller band parameters $\gamma_2$, $\gamma_5$ and $\delta$ a little over the known values for bulk graphite[19] to understand our experimental data (see Supplementary Table 1 and Supplementary Note 3 for estimation of band parameters from experimental LL crossing points). We find $\gamma_0 = 3.1 \, eV$, $\gamma_1 = 0.39 \, eV$, $\gamma_2 = -0.028 \, eV$, $\gamma_5 = 0.01 \, eV$ and $\delta = 0.021 \, eV$ best describe our data. Figure 3b shows the main fan diagram where the measured longitudinal conductance ($G_{xx}$) is plotted as a function of $V_{bg}$ and **B**. Due to lack of inversion symmetry, valley degeneracy is not protected in ABA-TLG, it breaks up with increasing **B** and reveals all the symmetry broken filling factors as seen in Fig. 3b.

Figure 3c shows measured $G_{xx}$ focusing on the $\nu = 0$ state, which shows a dip right at the charge neutrality point, evident for **B** > 6 T. Corresponding Hall conductance ($G_{xy}$) shows a plateau at zero indicating the occurrence of the $\nu = 0$ state (see Supplementary Fig. 4 and Supplementary Note 4 for longitudinal and Hall resistance data showing $\nu = 0$ state). While, the $\nu = 0$ plateau has been observed in monolayer graphene[20] and in bilayer graphene[21] (for **B** more than $\sim 15$–25 T), this is the first observation of $\nu = 0$ state in trilayer graphene at such low **B**. A marked reduction in disorder allows observation of the $\nu = 0$ state in our device.

Focusing on the electron side, Fig. 3d,e show the experimentally measured LL fan diagram and labelled LLs, respectively. We see that the presence of $N_M = 0$ LL gives rise to a series of vertical crossings along the **B** axis as is expected from the LL energy diagram (Fig. 3a). The highest crossing along the **B** axis appears when $N_M = 0$ crosses with $N_B = 2$ LL at $\sim 5$ T.

From the complex fan diagram, seen in Fig. 3d,e, we can see both above and below the topmost LL crossing ($V_{bg} \sim 10$ V and **B** $\sim 5$ T), $N_M = 0$ LL is completely symmetry broken and $N_B = 2$ LL quartet, on the other hand, becomes two-fold split at $\sim 3.5$ T. The crossing between $N_M = 0$ and $N_B = 2$ LLs gives rise to three ring-like structures. Calculated LL energy spectra near the topmost crossing (Fig. 3e, inset) shows that spin splitting is

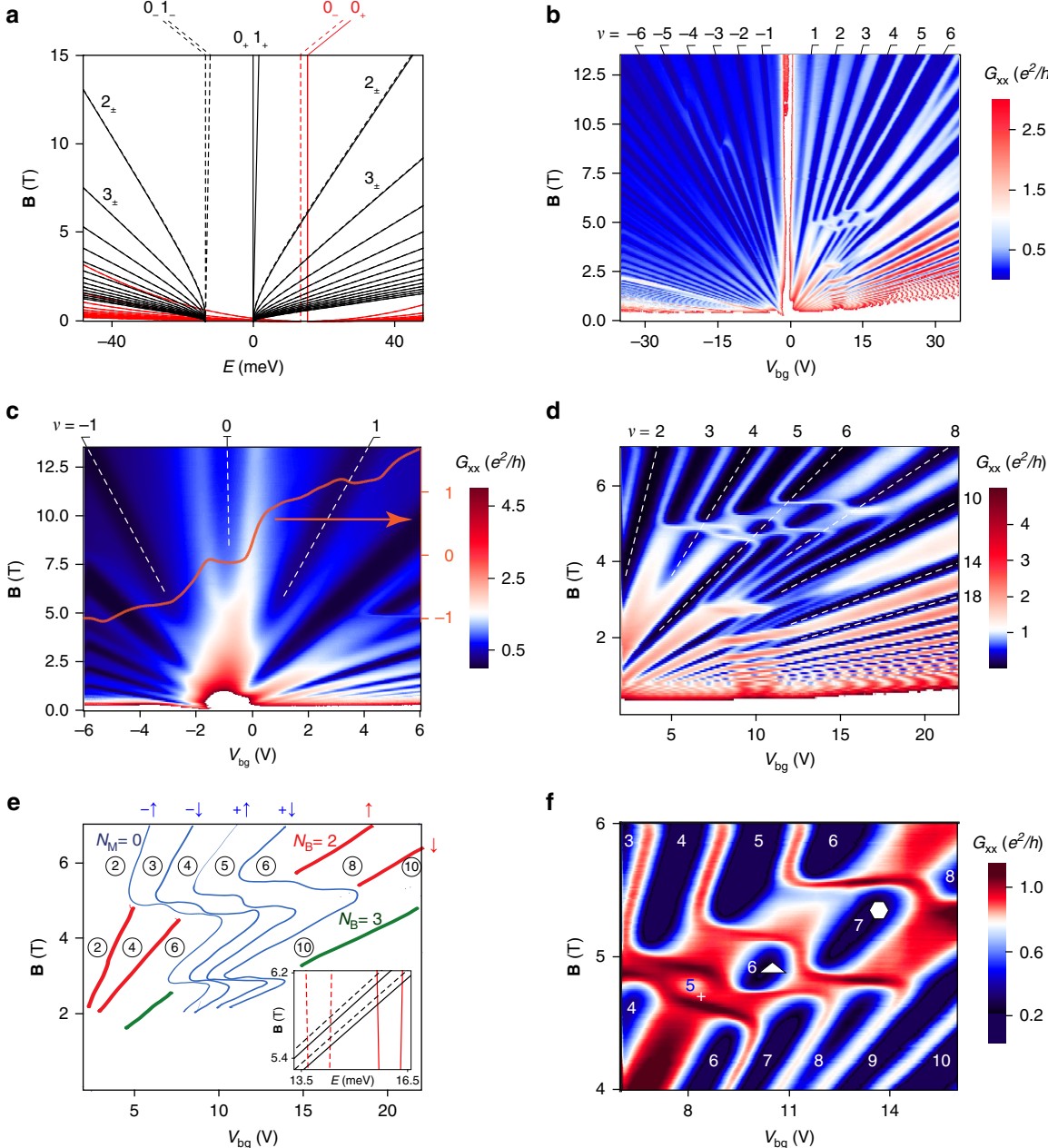

**Figure 3 | LL crossings and resulting QHF ground states.** (**a**) Calculated low energy spectra using SWMcC parametrization of the tight binding model for ABA-TLG[14–16]. Red and black lines denote the ML and BL LLs respectively. Solid and dashed lines denote LLs coming from $k_+$ and $k_-$ valleys respectively. Labelled numbers represent the LL indices of the corresponding LLs. (**b**) Colour scale plot of $G_{xx}$, showing the LL fan diagram. $v = 0$ feature is not seen in this colour scale as the lock-in sensitivity was set to a low value in this measurement to record the low resistance values accurately. The filling factors measured independently from the $G_{xy}$ are labelled in every plot. As a function of the **B**, one can observe several crossings on electron and hole side. The data shown in Fig. 2a forms a very thin slice of the low **B** data shown in this panel. (**c**) Zoomed-in fan diagram around charge neutrality point showing the occurrence of $v = 0$ from $\sim 6$ T: $G_{xx}$ shows a dip and $G_{xy}$ shows a plateau at $v = 0$. The overlaid red line presents $G_{xy}$ at 13.5 T which shows the occurrence of $v = -1$, 0 and 1 plateaus. (**d**) Zoomed-in recurrent crossings of $N_M = 0$ LL with different BL LLs. (**e**) The lines indicate the LLs seen in the data shown in **d** and their crossings. Circled numbers denote the filling factors. (**f**) A further zoomed-in view of the parameter space showing LL crossing of fourfold symmetry broken $N_M = 0$ LL with spin split $N_B = 2$ LL.

larger than valley splitting for $N_B = 2$ LL but valley splitting dominates over spin splitting for $N_M = 0$ LL. We note that valley splitting of $N_M = 0$ is very large compared with other ML LLs; which arises because ML bands are gapped in ABA-TLG unlike in monolayer graphene. As one follows the $N_M = 0$ LL down towards **B** $= 0$ one observes successive LL crossings of $N_M = 0$ with $N_M = 2, 3, 4.....$. The sharp abrupt bends in the fan diagram occur due to the change of the order of filling up of LLs after

crossings and the fact that the horizontal axis is charge density (proportional to $V_{bg}$ and not LL energy). When these crossings are extrapolated to **B** $= 0$, we see that $N_M = 0$ LL is valley split as expected from the LL energy diagram Fig. 3a.

**Role of electronic interaction and theoretical simulation.** We next discuss experimental signatures that point towards the importance of interaction. Observation of spin split $N_M = 0$ LL at

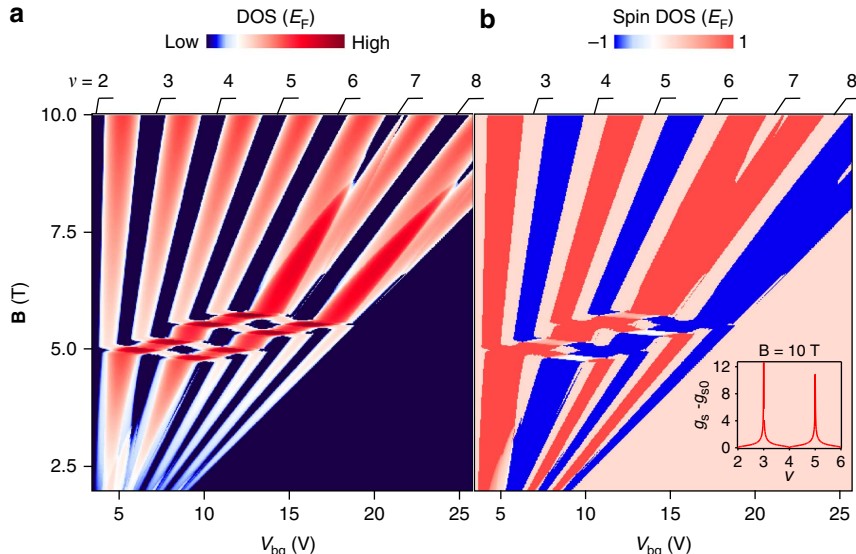

**Figure 4 | Theoretical calculation of DOS and spin polarization.** (**a**) DOS at the Fermi level as a function of $V_{bg}$ and **B**. This matches the fan diagram seen in the experiment. (**b**) The magnetization in the system as a function of $V_{bg}$ and **B**, where density is converted to an equivalent $V_{bg}$, described in Fig. 2b caption. The LL crossing regions clearly show the presence of spin polarization in the system. The inset shows calculated enhanced spin $g$-factor above the bare value 2 for $N_M = 0$ spin and valley split LLs.

**B** $\sim 2$ T cannot be explained from the non-interacting Zeeman splitting for $\Gamma \sim 1.5$ meV on electron side, estimated from the Dingle plot. Also, the large ratio of transport scattering time ($\tau_t$) to quantum scattering time ($\tau_q$)($\frac{\tau_t}{\tau_q} \approx 49$) indicates that small angle scattering is dominant, a signature of the long-range nature of the Coulomb potential[22–24] (see Supplementary Fig. 5 and Supplementary Note 5 for Dingle plot analysis). We also measure activation gap for the symmetry broken states $v = 2, 3, 4, 5, 7$ at **B** $= 13.5$ T, and find significantly higher gaps than the non-interacting spin-splitting. For $v = 3$ and 5, Fermi energy ($E_F$) lies in spin-polarized gap of $N_M = 0$ LL in $K_-$ and $K_+$ valley respectively. Measured energy gap at $v = 3$ is $\sim 5.1$ meV and at $v = 5$ is $\sim 2.8$ meV, whereas free electron Zeeman splitting is $\sim 1.56$ meV at **B** $= 13.5$ T (see Supplementary Fig. 6, Supplementary Table 2 and Supplementary Note 6 for determination of LL energy gaps from Arrhenius plots). We note that typically the transport gap tends to underestimate the real gap due to the LL broadening, so actual single particle gap might be even larger. This shows the clear role of interactions even with a conservative estimate of the LL gap.

Interaction results in symmetry broken states at low **B** that are QHF states. For the data in Fig. 3d, $v = 2, 3, 4, 5$ are QHF states for **B** $> 5.5$ T. Similarly, $v = 7, 8, 9$ are also QHF states for $5.5$ T $>$ **B** $> 4$ T. In fact the LLs associated with $v = 3, 4, 5$ after crossing are the same ML LLs which are responsible for $v = 7, 8, 9$ before crossing (Fig. 3e). The crossings result in three ring-like structures marked by plus, triangle and hexagon in Fig. 3f.

Now we discuss theoretical calculations to show that electronic interactions are crucial in obtaining a quantitative understanding of the experimental data. The theoretical calculations focus on the $N_M = 0$ and $N_B = 2$ LLs, which form the most prominent LL crossing pattern in our data. The effect of disorder is incorporated within a self-consistent Born approximation (SCBA)[25,26], while electronic interactions are included by considering the exchange corrections to the LL spectrum due to a statically screened Coulomb interaction[27,28] in a self-consistent way. Figure 4a shows the DOS at $E_F$ as a function of $V_{bg}$ and **B**, which matches with the experimental results on the $G_{xx}$.

Our calculations also provide insight about the polarization of the states inside the ring-like structures (Fig. 3f). We find that although the filling factor of region $\Delta$ is the same as that of regions $v = 6$ above and below, electronic configurations of these states are different. Figure 4b shows the spin-resolved DOS at $E_F$ as a function of $V_{bg}$ and **B**. We find total spin polarization (integrated spin DOS) in region $\Delta$ is non-zero (see Supplementary Fig. 7 and Supplementary Note 7 for the details of theoretical calculation), but it vanishes in regions $v = 6$ above and below the ring structure. Figure 4b inset shows the calculated exchange enhanced spin $g$-factors. This shows a significant increase over bare value of $g$ in the spin-polarized states—in agreement with the large gap observed at $v = 3$ and 5 in the experiment.

## Discussion

The key role of interactions is also reflected in the hysteresis of $R_{xx}$ in the vicinity of the symmetry broken QHF states. Though QHF has been extensively studied in 2DES using semiconductors[2,29] there are only a few reports of studying QHF in graphene[30–33]. In our experiment, we vary filling factor by changing $V_{bg}$ at a fixed **B** (Fig. 5a) and observe that the sweep up and down of $V_{bg}$ shows a hysteresis in $R_{xx}$, which can be attributed to the occurrence of pseudospin magnetic order at the symmetry broken filling factors[34] (see Supplementary Fig. 8 and Supplementary Note 8 for more hysteresis data). Corresponding hysteresis is absent in simultaneously measured Hall resistance $R_{xy}$ (Fig. 5b). Hysteresis in $R_{xx}$ with $V_{bg}$ is also absent without magnetic field (see Supplementary Fig. 8 and Supplementary Note 8 for more detail). The pinning, that causes the hysteresis could be due to residual disorder within the system as the domains of the QHF evolve. Along a constant filling factor $v = 6$ line (Fig. 5c) transport measurements show an appearance of $R_{xx}$ spikes around the crossing of $N_M = 0$ and $N_B = 2$ LLs (Fig. 5d). One possible explanation for the spike in $R_{xx}$ (ref. 35) is the edge state transport along domain wall boundaries as studied earlier in semiconductors[29,36].

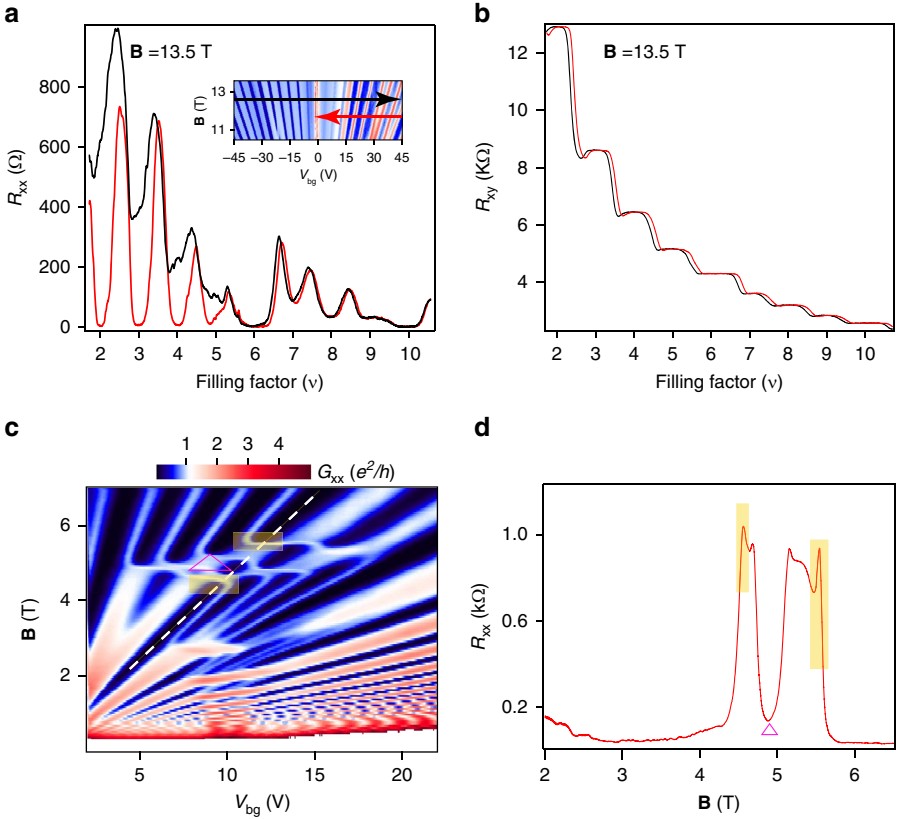

**Figure 5 | Hysteresis in the longitudinal resistance as a function of the filling factor and observation of resistance spikes.** (**a**) Measurement of $R_{xx}$ as a function of $V_{bg}$ at $\mathbf{B} = 13.5$ T in the two directions as shown in (inset) the measurement parameter space. Largest hysteresis is seen for the spin and valley-polarized $N_M = 0$ LL. We have done gate sweep as slow as $3\,mV\,s^{-1}$ to check if the hysteresis goes away, however, it stays. Nature of hysteresis does not change. The sweep up and sweep down rates were same. (**b**) Simultaneously measured $R_{xy}$ that exhibits clear quantization plateaus in the two sweep directions. (**c**) The laid white dashed line on the fan diagram shows the parameter space along which the $R_{xx}$ is plotted in the next panel. (**d**) $R_{xx}$ plotted along the dashed line shown in the parameter space. Spikes in resistance, shaded in yellow, correspond to boundaries of the region marked $\Delta$ in **c**.

In summary, we see interaction plays an important role to enhance the g-factor and favours the formation of QHF states at low **B** and at relatively higher temperature. ABA-TLG is the simplest system that has both massless and massive Dirac fermions, giving rise to an intricate and rich pattern of LLs that, through their crossings, can allow a detailed study of the effect of interaction at sufficiently low temperature. The ability to image these QHF states using modern scanning probe techniques at low magnetic fields could provide insight into these states that have never been imaged previously. In future, experiments on multilayer graphene, exchange coupled with a ferromagnetic insulating substrate[37], can lead to the possibility of observing an exciting interplay of QHF with the proximity induced ferromagnetic order.

## Methods

**Device fabrication.** Graphene and hBN flakes were exfoliated by scotch tape method on 300 nm $SiO_2$ coated highly p-doped Si substrate. hBN flakes of thickness $\sim 30$ nm were located by an optical microscope. ABA-TLG was then transferred to a suitably chosen hBN, followed by another hBN transfer of similar thickness to complete the hBN-graphene-hBN stack. Electron-beam lithography was done to define the contacts. Then the stack was etched with mild plasma in Argon and Oxygen (1:1 ratio) environment to expose the graphene edge. Metal (3 nm Chromium, 15 nm Palladium, 30 nm Gold) was thermally evaporated to make the contacts immediately after etching without breaking the vacuum.

**Characterization.** After exfoliation on Silicon substrate potential ABA-TLG graphene flakes were chosen by the optical colour contrast and then confirmed by the Raman spectroscopy. Atomic force microscopy was also done on the complete

stack to image the topography of the surface and to find out the thickness of the top hBN which is required to calculate the plasma etching time before metallization.

**Measurement.** All the low-temperature measurements were done in a liquid $He_4$ flow cryostat at base temperature $T = 1.5$ K. Standard low frequency lock-in technique was used to do all current biased four-probe resistance measurements. Excitation current was 100 nA for most of the measurements but sometimes increased to a higher value of 400 nA to measure low resistances at low magnetic fields.

**Data availability.** The data that support the findings of this study are available from the corresponding author upon request.

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

## Acknowledgements

We thank Allan MacDonald, Jainendra Jain, Jim Eisenstein, Fengcheng Wu, Vibhor Singh, Shamashis Sengupta and Chandni U. for discussions and comments on the manuscript. We also thank John Mathew, Sameer Grover and Vishakha Gupta for experimental assistance. We acknowledge Swarnajayanthi Fellowship of Department of Science and Technology (for M.M.D.) and Department of Atomic Energy of Government of India for support. Preparation of hBN single crystals is supported by the Elemental Strategy Initiative conducted by the MEXT, Japan and a Grant-in-Aid for Scientific Research on Innovative Areas 'Science of Atomic Layers' from JSPS.

## Author contributions

B.D. fabricated the device, conceived the experiments and analysed the data. M.M.D., A.B. and B.D. contributed to the development of the device fabrication process. H.A. helped in the fabrication and in the measurements. K.W. and T.T. grew the hBN crystals. S.D., A.S. and B.D. did the calculations under the supervision of R.S.; B.D., M.M.D. co-wrote the manuscript and R.S. provided input on the manuscript. All authors commented on the manuscript. M.M.D. supervised the project.

## Additional information

**Competing financial interests:** The authors declare no competing financial interests.

**Publisher's note**: 

