## [Peer Review File · Nature Communications]

Reviewers' comments:

Reviewer #1 (Remarks to the Author):

In this paper the authors report on the study of LL spectrum in single gated ABA trilayer graphene. They argue that the Landau Fan diagram indicates that the electron-electron interactions lead to formation of broken symmetry spin polarized states. The authors argue this by comparing the measured spin-polarized gaps with those theoretically predicted by pure Zeeman coupling.

The paper is a nice study and makes physical sense, but I don't believe the physics discussed here is novel enough to merit publication in Nature Communications. The physics of quantum Hall ferromagnetism (QHF) has been well known and has been studied for the past decade in semiconductors and 2D crystals. In my opinion, the observation of QHF in another system is not surprising or novel enough. The major claims of this paper that interactions can lead to spontaneously broken symmetries is not that profound or groundbreaking. I think the paper should be published in a more technical journal. In addition, I have some comments and questions on the current manuscript.

1) The SWMcC parameters for graphite used by the authors to fit the Landau fan diagram maybe different from those of ABA trilayer graphene. Why not determine these parameters by fitting to their experimental data?

2) The authors have a single gated sample but they argue that they have zero perpendicular electric field in their samples. This is not entirely true as this approximation is limited by the experimental resolution of their data (LL broadening). I think the authors need to discuss the lower bound they can establish on the electric field, and discuss its effects.

3) The authors need to fix some grammatical errors I noticed while reviewing the MS.

Again, I think the paper is nice and should be published, but I don't think the main physics is sufficiently novel for publication in Nature Communications.

Reviewer #2 (Remarks to the Author):

In the manuscript by Professor Deshmukh and co-workers entitled "Strong electronic interaction and multiple quantum Hall ferromagnetic phases in trilayer graphene" magneto-transport on single-gated, high mobility, BN encapsulated, ABA-graphene trilayer have been reported. The Landau level fan diagram shows complete degeneracy lifting of the 12-fold zeroth Landau level and clear LL level crossing confirming the ABA nature of the sample. The crossing between of the N=0 monolayer like Landau level and the N=2 bilayer like Landau level give rise to ring like structures. These structures are interpreted, in the absence of electric displacement field, as signature of long-range Coulomb interactions.

In a single-particle picture when no electric displacement field is applied between the top and bottom layer, no hybridization between the MLG-like and the BLG-like bands occurs. Experimental results obtain on dual gated trilayer graphene have confirmed the validity of this hypothesis (see e.g. PRL 117, 066601(2106), Nano lett. 13, 1627(2013), Nature phys 7, 621 (2011)). The submitted manuscript does a good job convincing that Coulomb interactions play a role in the observed magneto-transport measurement. The simulation taking into account electronic interactions match the data quite well. The presence of hysteresis in the transport curves for given filling factor states give indications of the QHF nature of the symmetry broken state.

1. My major point of concern is that unfortunately, the sample is single gated so it does not allow

the independent control of the carrier density n , the electric displacement field, and the potential of the middle layer. These three parameters have been proved to be of prime importance on the band hybridization (PRB 87, 115422 (2013)) and will be strongly affected (in a complicated way) by application of a back gate voltage. The authors try to answer problem by simulating the expected behavior for several value of $\Delta 1$. Due to screening, the potential of each layer would need to be recalculated for every carrier density, considering this I don't believe that the zero electric field approximation used there after in the manuscript is satisfactory.

I think that without top gate this work have a hard time to compare previous work on this subject or parallel domain of studies like ABC-trilayer graphene (Nano lett. 16, 227 (2016), Nature Comm. 5, 6656 (2014)) which all allow to study the competition between single particle and many body physics.

Some other points would need clarification:

2. Figure 1c does not give useful information about the sample: What is the sample geometry? What is the aspect ratio?

3. At the charge neutrality point, the resistivity of the sample reach 200 Ohm, how do you explain such a low resistivity? (Compared to what have been previously observed ? PRL 117, 066601(2106), Nano lett. 13, 1627(2013), Nature phys 7, 621 (2011)). I would expect the resistivity to go up at the CNP when decreasing the temperature. (Figure 1 d)

4. Figure 5, the critical parameter concerning the hysteresis observed in R_{xx} at the vicinity the QHF is the magnetic field. It would therefore be conceptually easier to understand the experimental data on hysteresis if plotted versus B and at constant V_g (or v).

5. If the presence of hysteresis is correlated to the suppression of the affected v states, it is not clear for me why only the longitudinal resistance (R_{xx}) is affected and not the Hall effect (R_{xy}).

6. I think this study of hysteresis lack a study of the temperature dependence, which would give information on the energy scale involved and confirm or infirm is relation with electron-electron interactions.

7. The manuscript only refer to measurement on one sample, does the authors have observed similar transport behavior in other samples ?

In conclusion, in my opinion the submitted work, even if it presents interesting new phenomenon, is incomplete compare to previous works published in comparable reviews. Therefore I do not recommend the publication of this manuscript in Nature Communications

Reviewer #3 (Remarks to the Author):

This manuscript reports on magneto-transport studies of ABA-stacked trilayer graphene as a function of charge-carrier density and magnetic field up to 13.5T. This work enhances the understanding of quantum Hall effect (QHE) in multilayers graphene by explaining the importance of the electron-electron interactions in the transition between different classes of Landau Levels whose position depends on interlayer potential. The authors perform a number of theoretical calculations and experiments to prove the significance of Coulomb interactions in the experimental data. In addition, authors observe hysteresis in the experimental data while sweeping the charge carrier density up and down on the electron side of the Landau fan diagram.

In the first part of the manuscript authors are able to demonstrate a good agreement with the theoretical predictions at relatively low magnetic fields. In the second part authors consider the broken-symmetry states at low fillings attributed to QHF. This part is less convincing and requires more insights for the final version of the manuscript.

A number of questions that need to be addressed before publication in Nature Communications:

1) Page 3, paragraph 2. Authors use SWMcC parameters for tight-binding model and claim to use all of them to describe the present data. However, there is no information about what values of γ_3 and γ_4 parameters were chosen for parameterization.

2) Page 3, paragraph 3. Authors present the evidence for the $\nu = 0$ QHS in trilayer graphene and find that there is a prominent dip in G_{xx} data while G_{xy} data reveals a plateau at zero-charge carrier density. This behavior of G_{xx} has not been observed in monolayer/bilayer graphene studies that have been cited in the main text. This feature of G_{xx} at CNP can indicate some additional phenomena different from those in mono/bilayer graphene. Authors do not present R_{xy} data for the CNP neither in supplementary materials nor in the main text, which makes it hard to check and requires more detailed explanations.

3) Fig. 3b doesn't show similar dip at $\nu = 0$ that is present on the Fig. 3c. However, it might be a color scale issue. Please, address this inconsistency.

4) Page 4, paragraph 1. Authors claim that "abrupt bends in the fan diagram occur due to the change of the order of filling" which implied the presence of the phase transition, i.e. monolayer-like band becomes bilayer-like band with the same filling before and after the crossings. Are these "bends" the same ones showed as spikes in resistance on the Fig. 5b where they claimed to be signatures of QHF in TLG? In this case, why does it exhibit abrupt change in the resistance?

5) Page 4, paragraph 2. Temperature range used for extracting of the activation gaps is very narrow and varies between 25-31K for $\nu=2$, 7-16K for $\nu=3$, 12-16K for $\nu=4$, 7-14K for $\nu=5$ and 6-10K $\nu=7$. Besides being quiet small ranges and away from the base temperature 1.5K, this must introduce additional error in the final calculations due to thermal broadening of LL, which is the order of 0.5-3 meV. Is it included in the total error?

Let's now consider QHS with $\nu=4$ (attributed to $NM = 0(-)$) and $\nu=5$ (attributed to $NM = 0(+)$). According to the Arrhenius plot they have gaps of 4.37 and 2.8 meV's, respectively, at $B = 13.5T$. This implies that QHS with $\nu=4$ has to demonstrate a crossing below $B = 13T$ since the valley-split $NM = 0(-)$ and $NM = 0(+)$ have a separation of $\sim 2-3$ meV according to Fig. 3a. However, this crossing doesn't appear on the Fig. 3b. In other words, according to the Arrhenius plot, interactions have to push symmetry-broken LL's to close a single-particle gap between $NM = 0(-)$ and $NM = 0(+)$ but there is no any experimental evidences. Does it deal with valley ferromagnetism or there is some other mechanisms involved?

There is another feature that authors do not discuss in the main text, even though it is interesting. For $\nu = -4$, one can observe a clear crossing feature at $\sim 9.5T$. QHS with $\nu=-4$ is formed between $NB = 0(-)$ and $NB = 1(-)$ and there is no prediction about this crossing in calculated LL fan diagram in the Fig. 3a. In Fig. 3a the gap size between these LL's is $\sim 1-2$ meV at 10T and free electron Zeeman splitting at this magnetic field is ~ 1.1 meV ($g = 2$) which could explain a crossings at $\nu = -4$. What is the role of interactions on the hole-side?

Another possible scenario might be that there is some electric field potential across the sample that leads to the gap closure. The justification of the zero-electric field approximation described in the supplementary materials is not convincing. Authors claim that in case of electric field presence there should be some "signature of anti-crossing in experiment". What type of "signature" could be observed in experiment? The justification of this statement might include some tilted magnetic field measurements. However, authors do not include them.

Please, address this questions.

6) Page 5, paragraph 3. No information is given about experimental details: how fast was the gate swept up and down (were the speeds the same for sweep up and down)? Did authors try to wait at the highest/lowest back gate voltages for some sufficient enough time to see if the hysteresis signatures preserve?

We thank the reviewers for their comments, they have helped to make the manuscript stronger.

Point by point response to reviewers' comments

(Reviewers' questions are in blue coloured italic font and our responses are in black coloured font)

Reviewer #1 (Remarks to the Author):

In this paper the authors report on the study of LL spectrum in single gated ABA trilayer graphene. They argue that the Landau Fan diagram indicates that the electron-electron interactions lead to formation of broken symmetry spin polarized states. The authors argue this by comparing the measured spin-polarized gaps with those theoretically predicted by pure Zeeman coupling.

We thank the reviewer for noting "The paper is a nice study and makes physical sense ..."

I have some comments and questions on the current manuscript.

- 1. The SWMcC parameters for graphite used by the authors to fit the Landau fan diagram maybe different from those of ABA trilayer graphene. Why not determine these parameters by fitting to their experimental data?*

As correctly pointed out by the reviewer the experimental Landau level crossing points can be used to determine the hopping parameters. Here we have used the crossing points of monolayer-like $N_M=1$ Landau level (LL) with other bilayer-like Landau levels from LL index $N_B=17$ to $N_B=26$ to calculate different hopping parameters. Since, γ_0 and γ_1 are known very precisely, we vary relatively smaller hopping parameters γ_2 , γ_5 and δ to match the experimentally observed magnetic field values at the crossing points.

Following are the comparison of the experimental and theoretical crossing points (with the hopping parameters mentioned in the main text). Experimentally Landau level orbital index of the Landau levels is determined by the help of filling factors determined from the experimental Hall conductance (G_{xy}) plot.

LL index of monolayer-like LLs (N_M)	LL index of bilayer-like LLs (N_B)	Experimental magnetic field (T) at crossing point	Theoretical magnetic field (T) at crossing point
1	17	1.47	1.47
1	18	1.35	1.35
1	19	1.24	1.24
1	20	1.14	1.15
1	21	1.05	1.06
1	22	0.98	0.99
1	23	0.92	0.92
1	24	0.86	0.87
1	25	0.80	0.82
1	26	0.76	0.77

While we had calculated band parameters using these observations in the last version of the draft (main as well as Supporting Information) we did not emphasize this as this has been measured before by others (Nature Physics 7, 621 (2011)).

We have now included this table in the supporting information to make this aspect clear.

- 2. The authors have a single gated sample but they argue that they have zero perpendicular electric field in their samples. This is not entirely true as this approximation is limited by the experimental resolution of their data (LL broadening). I think the authors need to discuss the lower bound they can establish on the electric field, and discuss its effects.*

In order to address the queries of the reviewer, we have now done measurements on another dual-gated ABA-trilayer graphene device which suggests that at zero electric field the LL crossings on electron side in the fan diagram doesn't change the conclusions and analysis presented. Hole side crossings are relatively more sensitive to the electric field compared to the electron side. Our manuscript focused on studying the physics on the LL crossings on the electron side.

Our new measurements on the dual-gated ABA-trilayer sample done at various electric fields suggest the presence of maximum stray electric field ~ 0.01 V/nm in the earlier single-gated measurements. While our experimental results were consistent in the previous version, our new measurements justify our use of $E=0$ V/nm in the theoretical calculations presented.

Figure 1: Data from new dual gated device

3. *The authors need to fix some grammatical errors I noticed while reviewing the MS. Again, I think the paper is nice and should be published, but I don't think the main physics is sufficiently novel for publication in Nature Communications.*

We have reread the manuscript and made a conscious effort to fix the grammatical errors. We thank the reviewer for mentioning this.

We respect the reviewer's concern and understand that there already have been a lot of novel studies done on quantum Hall ferromagnetism (QHF) in GaAs-AlGaAs heterostructure based 2DEG; however, the experimental studies of QHF in graphene based systems are very few (Nature Physics 8, 550 (2012), Science 345, 58 (2014)). Our work is dedicated to the study of QHF in trilayer graphene. We mention a few points which show the novel aspects of our work:

- Studying QHF in graphene is very interesting due to the high energy scale involved which enables us to probe the QHF physics up to much higher temperature (~ 10 K).
- For the same reason, the involved magnetic field can be quite low (~ 3 T) which opens up the possibility of doing many other measurements on these systems including imaging these QHF states using modern scanning probe techniques.
- There are only a few experimental reports on the QHF states in monolayer graphene (Nature Physics 8, 550 (2012)) at very high magnetic fields ~ 15 - 35 T and in bilayer graphene (Science 345, 58 (2014)). Improved mobility enables us to probe a large number of QHF states in a more interesting trilayer graphene system at much lower magnetic field.
- Our data quality is distinctly improved compared to the other experiments including two most recent trilayer graphene works (PRL 117, 076807 (2016), PRL 117, 066601 (2016)). We see much more variety of symmetry broken states in the fan diagram at a similar magnetic field and temperature which we explain taking electronic interaction into account. We have also added a table below showing the comparison of mobility from various recent monolayer (MLG), bilayer (BLG) and trilayer graphene (TLG) experiments.

Citation	Sample	Mobility (μ) ($\text{cm}^2\text{V}^{-1}\text{s}^{-1}$)
Our study	ABA TLG	~ 500,000 (electron) and 800,000 (hole) at 1.5 K
Ref. 4 (Nature Physics 7, 621 (2011))	TLG	110,000 at 300 mK, 65,000 at 40 K
Ref. 5 (Nature Physics 7, 948 (2011))	BLG, TLG	210 to 1,900 at 4K
Ref. 6 (Physical Review Letters 107, 126806 (2011))	MLG, BLG, TLG	1,200 at 4.2 K
Ref. 8 (Physical Review X 2, 011004 (2012))	ABA TLG	4,000 at 0.3 K
Ref. 9 (Nature nanotechnology 4, 383 (2009))	TLG	800 at 4.2 K
Ref. 11 (Nano letters 13, 1627 (2013))	TLG	15,000 at 300 mK
Ref. 12 (Physical Review Letters 117, 076807 (2016))	ABA TLG	100,000 at 260 mK
Ref. 13 (Nature Communications 5, 5656 (2014))	ABC TLG	20,000 to 90,000 at 260 mK
Ref. 31 (Nature Physics 8, 550 (2012))	MLG	491.94 at 2K (Calculated from σ vs V_g plot)
Ref. 32 (Science 345, 58 (2014))	BLG	100,000 to 290,000 at 4K
Ref. 33 (Nano letters 16, 227 (2016))	ABC TLG	42,000 at 270 mK

Reviewer #2 (Remarks to the Author):

We thank the reviewer for noting “The submitted manuscript does a good job convincing that Coulomb interactions play a role in the observed magneto-transport measurement. The simulation taking into account electronic interactions match the data quite well. The presence of hysteresis in the transport curves for given filling factor states give indications of the QHF nature of the symmetry broken state. “

1. *My major point of concern is that unfortunately, the sample is single gated so it does not allow the independent control of the carrier density n , the electric displacement field, and the potential of the middle layer. These three parameters have been proved to be of prime importance on the band hybridization (PRB 87, 115422 (2013)) and will be strongly affected (in a complicated way) by application of a back gate voltage. The authors try to answer problem by simulating the expected behavior for several value of $\Delta 1$. Due to screening, the potential of each layer would need to be recalculated for every carrier density, considering this I don't believe that the zero electric field approximation used there after in the manuscript is satisfactory.*

I think that without top gate this work have a hard time to compare previous work on this subject or parallel domain of studies like ABC-trilayer graphene (Nano lett. 16, 227 (2016), Nature Comm. 5, 6656 (2014)) which all allow to study the competition between single particle and many body physics.

To address the reviewer's concern about the finite stray electric field we have fabricated a device with both top and bottom gates. The new measurements on this device suggest the presence of maximum 0.01 V/nm electric field in the earlier measurements of our previous version of the manuscript but that does not affect any of our conclusions. The presence of a small electric field has a little effect on the Landau level crossing of $N_M=0$ and $N_B=2$ on electron side. We have included the new measurement data at zero and 0.01 V/nm electric field below and also in the Supplementary Information.

Figure 2: Data from new dual gate device

Some other points would need clarification:

2. *Figure 1c does not give useful information about the sample: What is the sample geometry? What is the aspect ratio?*

Figure 3: Optical image of the hBN encapsulated trilayer graphene device

The graphene sample is a slightly distorted rectangle, but the electrodes are designed in a Hall bar geometry. Length and breadthwise distance between furthest electrodes are 9.3 μm and 7.8 μm respectively. This makes the aspect ratio to be 1.19. The dashed line in the figure marked the extent of the graphene flake. We now explicitly mention this in the caption of the figure.

We want to mention that the edges of graphene are not sculpted into a Hall bar geometry, as we believe that introduces disorder. We form the edge contacts by etching inside the evaporator and depositing contacts inside without breaking the vacuum. Our modified fabrication technique we believe is the reason for the cleanliness of our devices and the high mobility we measure.

3. At the charge neutrality point, the resistivity of the sample reach 200 Ohm, how do you explain such a low resistivity? (Compared to what have been previously observed? PRL 117, 066601(2106), Nano lett. 13, 1627(2013), Nature phys 7, 621 (2011)). I would expect the resistivity to go up at the CNP when decreasing the temperature. (Figure 1 d)

Figure 4: Temperature dependence of the resistance

ABA-trilayer graphene is metallic due to the band overlap and hence density of states at the “zero” energy (Fermi energy) never vanishes, so, theoretically a low resistance at the charge neutrality point can be expected. In fact, there is always some finite density of states at the charge neutrality point due to the monolayer-like bands even when the bilayer-like bands are gapped out.

However, as we know, due to the contact resistance experimentally the measured device resistance depends on the quality of metal contact. The reported device in the manuscript has a very low contact resistance and hence we can measure the

device resistance very precisely. This, in turn, reflects on the quantum Hall measurements showing extraordinary clean data compared to the most recent data on similar systems (PRL 117, 076807 (2016), PRL 117, 066601 (2016)). We can get a rough estimate of the resistivity from the Drude formula $\rho = 1/(n e \mu)$. Plugging in the charge density $n = 6.5 \times 10^{14} \text{ m}^{-2}$ at $V_{bg} = 1 \text{ V}$, mobility $\mu \sim 80 \text{ m}^2 \text{ V}^{-1} \text{ s}^{-1}$ and electronic charge $e = 1.602 \times 10^{-19} \text{ C}$ we get $\rho = 120 \text{ } \Omega$ which is close to the experimentally measured resistance.

To answer the second part of the question, we show the resistivity of the device at an intermediate temperature 50 K. Our observation is that the resistance shows metallic behaviour from room temperature to around 30 K – resistance at charge neutrality point decreases with decreasing temperature. However, there is a crossover near 30 K, below 30 K it shows semiconducting behaviour – resistance increases with decreasing temperature.

4. Figure 5, the critical parameter concerning the hysteresis observed in R_{xx} at the vicinity the QHF is the magnetic field. It would therefore be conceptually easier to understand the experimental data on hysteresis if plotted versus B and at constant V_g (or v).

According to the reviewer’s suggestion, we have replotted the hysteresis data in the manuscript as a function of filling factor. Only a few symmetry broken Landau levels can be accessed sweeping the magnetic field at a particular gate voltage, hence we sweep gate voltage at a particular magnetic field so that we can access all the symmetry broken states during the hysteresis measurement.

5. *If the presence of hysteresis is correlated to the suppression of the affected ν states, it is not clear for me why only the longitudinal resistance (R_{xx}) is affected and not the Hall effect (R_{xy}).*

Longitudinal resistance R_{xx} depends on the bulk of the sample. Hence R_{xx} depends on the details of how domains are distributed spatially at the transition from one Landau level to another one. On the other hand, R_{xy} is insensitive to the bulk and only depends on the edge states which are robust to the backscattering. In other words, being a topological quantity, Hall resistance (R_{xy}) depends only on the number of extended states, not on the microscopic details of the quantum Hall ferromagnetic domains.

6. *I think this study of hysteresis lack a study of the temperature dependence, which would give information on the energy scale involved and confirm or infirm is relation with electron-electron interactions.*

We measured the temperature dependence of the hysteresis; the hysteresis persists nearly up to 10 K. Any measurement beyond 10 K results in poor resolution of the measured longitudinal resistance. So, unfortunately, we can give only the lower bound of the hysteresis associated energy scale which is ~ 10 K. We believe that this hysteresis energy scale provides an insight about the pinning and anisotropy, associated with the domain walls in the system rather than the electron-electron interaction.

7. *The manuscript only refer to measurement on one sample, does the authors have observed similar transport behavior in other samples?*

This is the best trilayer graphene sample which showed all the crossing features most clearly. We report all the data from one sample throughout the paper. In the Supporting Information, we have now included data from a double gated ABA-trilayer device which shows very similar physics. Needless to say the quality of the device in the main manuscript is substantially better.

Reviewer #3 (Remarks to the Author):

We thank the reviewer for noting “This work enhances the understanding of quantum Hall effect (QHE) in multilayers graphene by explaining the importance of the electron-electron interactions in the transition between different classes of Landau Levels whose position depends on interlayer potential. The authors perform a number of theoretical calculations and experiments to prove the significance of Coulomb interactions in the experimental data. In addition, authors observe hysteresis in the experimental data while sweeping the charge carrier density up and down on the electron side of the Landau fan diagram.”

1. *Page 3, paragraph 2. Authors use SWMcC parameters for tight-binding model and claim to use all of them to describe the present data. However, there is no information about what values of γ_3 and γ_4 parameters were chosen for parameterization.*

We can use all the hopping parameters (including γ_3 and γ_4) to determine the energy eigenvalues of the full trilayer graphene non-interacting Hamiltonian by numerical diagonalization. However, there exists a very good well-known approximation (PRL 96, 086805 (2006), PRB 81, 115315 (2010)) for low energy spectra which shows that an analytical solution of the non-interacting Hamiltonian is possible. Under this approximation higher order term containing γ_3 and γ_4 are neglected. Hence, first, we use this approximation to calculate the analytical wavefunctions of the non-interacting Hamiltonian and then use these wavefunctions to calculate the exchange energy correction self-consistently. Without the analytical wavefunctions the exchange energy calculation becomes very difficult and hence we use this approximation.

2. *Page 3, paragraph 3. Authors present the evidence for the $\nu = 0$ QHS in trilayer graphene and find that there is a prominent dip in G_{xx} data while G_{xy} data reveals a plateau at zero-charge carrier density. This behavior of G_{xx} has not been observed in monolayer/bilayer graphene studies that have been cited in the main text. This feature of G_{xx} at CNP can indicate some additional phenomena different from those in mono/bilayer graphene. Authors do not present R_{xy} data for the CNP neither in supplementary materials nor in the main text, which makes it hard to check and requires more detailed explanations.*

This is true that the cited papers (PRL 96, 136806 (2006), PRL 104, 066801 (2010)) do not show G_{xx} data. Our interpretation of their data suggests that their claim may be reasonable based on following arguments -- they show R_{xx} data which shows very high resistance (>100 K Ω) and G_{xy} data which shows zero conductance at the charge neutrality point. The presence of a plateau at zero conductance in G_{xy} would also mean a plateau at zero resistance in R_{xy} . We know longitudinal conductance is given by $G_{xx} = R_{xx} / (R_{xx}^2 + R_{xy}^2)$ which becomes $G_{xx} \sim 1/R_{xx}$ for $\nu = 0$ at the charge neutrality point. Since, R_{xx} shows a high peak, consequently,

G_{xx} shows a dip. Although in the cited papers authors do not show G_{xx} data, we expect that G_{xx} will reveal a dip at the $\nu = 0$ state.

To answer the second question we note that though we start resolving the $\nu = 0$ state from ~ 6 T, it is not completely developed even at our maximum attainable magnetic field 13.5 T. To address the reviewer's concern we have now added the R_{xy} data at 13.5 T magnetic field both here and in the new version of the Supplementary Information which shows a plateau at $R_{xy} = 0$.

Figure 5: Hall resistance plateaus

3. Fig. 3b doesn't show similar dip at $\nu = 0$ that is present on the Fig. 3c. However, it might be a color scale issue. Please, address this inconsistency.

This is true, the lock-in sensitivity was set to a low value in the measurement corresponding to Fig. 3b to record the low resistance values accurately and intentionally the region around $\nu=0$ was allowed to saturate. So, we could not show the correct value in this plot near $V_{bg} = 0$ ($\nu = 0$) which was done separately in Fig. 3c. So, the colour scale of Fig.3b did not reveal the feature at $V_{bg} = 0$. We hope that this addresses the concern of the reviewer. We have now added a note in the caption of the figure that clarifies this. We thank the reviewer for pointing this out.

4. Page 4, paragraph 1. Authors claim that "abrupt bends in the fan diagram occur due to the change of the order of filling" which implied the presence of the phase transition, i.e. monolayer-like band becomes bilayer-like band with the same filling before and after the crossings. Are these "bends" the same ones showed as spikes in resistance on the Fig. 5b where they claimed to be signatures of QHF in TLG? In this case, why does it exhibit abrupt change in the resistance?

Yes, these "bends" are the same ones which show spikes in the resistance.

These states are quantum Hall ferromagnetic states. Around the crossing of two different Landau levels sequence of the filling of electrons in the Landau levels changes – the character of the Landau level does not change. The monolayer-like band **does not** become the bilayer-like band.

Due to the different nature of ferromagnetism (valley or spin) they have different domains and dissipative electron scattering from the domain walls give rise to the resistance peaks. Similar resistance peaks were observed for the quantum Hall ferromagnetic states in 2DEG (PRL 87, 196801 (2001), Science 290, 1546 (2000)) which stems from the same physical phenomena.

5. Page 4, paragraph 2. Temperature range used for extracting of the activation gaps is very narrow and varies between 25-31K for $\nu=2$, 7-16K for $\nu=3$, 12-16K for $\nu=4$, 7-14K for $\nu=5$ and 6-10K $\nu=7$. Besides being quiet small ranges and away from the base temperature 1.5K, this must introduce additional error in the final calculations due to thermal broadening of LL, which is the order of 0.5-3 meV. Is it included in the total error?

The fits become a little poor when we extend our fitting range in the activation data. We also observe that the calculated LL gaps slightly decrease than the values we got from earlier relatively narrow range fits. We have included the new fits and a table summarizing the activation gaps from the new fits. We also note that the current fitting range is similar to what used in earlier works (Nature Physics 8, 550 (2012)).

Figure 6: New Arrhenius fits to determine LL gaps

Filling factor (ν)	Fitting range (K)	Activation gap ($\Delta_\nu - \Gamma$) (meV)
2	15-25	10.14 ± 0.06
3	7-16	4.1 ± 0.1
4	7-12.5	3.8 ± 0.1
5	7-14	2.4 ± 0.1
7	6-10	1.7 ± 0.1

To answer the second question, these errors don't account for the thermal broadening of the Landau levels. These are mere fitting errors. However, to find the true Landau level single particle gap we need to add up the disorder broadening (Γ) to it which is a temperature dependent quantity. Our Dingle plot analysis shows quantum broadening $\Gamma \sim 1-1.5$ meV at 1.5 K temperature which should increase with temperature. So, the lower bound of the actual activation gap is 1 meV more than the measured gap noted in the above table. This shows the clear role of interactions even with a conservative estimate of the Landau level gap.

6. Let's now consider QHS with $\nu=4$ (attributed to $N_M = 0(-)$) and $\nu=5$ (attributed to $N_M = 0(+)$). According to the Arrhenius plot they have gaps of 4.37 and 2.8 meV's, respectively, at $B = 13.5T$. This implies that QHS with $\nu=4$ has to demonstrate a crossing below $B = 13T$ since the valley-split $N_M = 0(-)$ and $N_M = 0(+)$ have a separation of $\sim 2-3$ meV according to Fig. 3a. However, this crossing doesn't appear on the Fig. 3b. In other words, according to the Arrhenius plot, interactions have to push symmetry-broken LL's to close a single-particle gap between $N_M = 0(-)$ and $N_M = 0(+)$ but there is no any experimental evidences. Does it deal with valley ferromagnetism or there is some other mechanisms involved?

From the Landau level mentioned by the reviewer, we think the reviewer means $\nu = 3$ state (not $\nu = 4$) which occurs between $N_M = 0-, \uparrow$ and $N_M = 0-, \downarrow$.

Yes, we believe that valley ferromagnetism is responsible for moving the $N_M = 0-$ and $N_M = 0+$ Landau level apart from each other, so that $N_M = 0-, \downarrow$ Landau level does not cross with $N_M = 0+, \uparrow$ Landau level, at least up

to 13.5 T. Fig. 3a shows the Landau levels without including any interaction, including interaction leads to increase in the valley gap. Fig. 4a shows the calculated fan diagram including the interaction which does not show crossing between $N_M=0^-$ and $N_M=0^+$ LLs, confirming that these two Landau levels do not cross each other in our observed magnetic field range.

7. *There is another feature that authors do not discuss in the main text, even though it is interesting. For $\nu = -4$, one can observe a clear crossing feature at $\sim 9.5T$. QHS with $\nu = -4$ is formed between $N_B = 0(-)$ and $N_B = 1(-)$ and there is no prediction about this crossing in calculated LL fan diagram in the Fig. 3a. In Fig. 3a the gap size between these LL's is $\sim 1-2meV$ at 10T and free electron Zeeman splitting at this magnetic field is $\sim 1.1meV$ ($g = 2$) which could explain a crossings at $\nu = -4$. What is the role of interactions on the hole-side?
Another possible scenario might be that there is some electric field potential across the sample that leads to the gap closure.*

Yes, as correctly pointed out by the reviewer that Zeeman splitting can lead to some additional crossings on the hole side. However, apart from that, we have observed that the hole side crossings are much more sensitive to the electric field than the electron side. Hole side crossings can be tuned with varying electric field, at zero electric field the crossings are absent. We have included the electron side data measured on a double gated ABA trilayer graphene device in the Supplementary Information.

The interplay between electronic interaction and effect of the electric field is more interesting on hole side. This can be an intriguing study and beyond the scope of our current manuscript, so, we plan to address it in detail in a separate study.

8. *The justification of the zero-electric field approximation described in the supplementary materials is not convincing. Authors claim that in case of electric field presence there should be some "signature of anti-crossing in experiment". What type of "signature" could be observed in experiment? The justification of this statement might include some tilted magnetic field measurements. However, authors do not include them. Please, address this questions.*

Now we have made a dual gate ABA-trilayer graphene device where we can independently vary the electric field. We see only little change on electron side at zero electric field. Our results are consistent with our reasonable approximation presented earlier.

The signature of anticrossing would be a drop in the conductance due to the low density of states at the anticrossing points.

Unfortunately, we do not have a way to do tilted field measurements at this moment.

9. *Page 5, paragraph 3. No information is given about experimental details: how fast was the gate swept up and down (were the speeds the same for sweep up and down)? Did authors try to wait at the highest/lowest back gate voltages for some sufficient enough time to see if the hysteresis signatures preserve?*

We have done gate sweep as slow as 3 mV/sec to check if the hysteresis goes away, however, it stays. Nature of hysteresis does not change. The sweep up and sweep down rates were same. We have now included these details in the manuscript.

We checked by waiting at the end points of back gate voltage, but the forward and backward sweeps across the quantum Hall ferromagnetic states always have hysteresis. However, two forward (backward) sweep resistances lie almost exactly on top of each other.

Reviewers' Comments:

Reviewer #1 (Remarks to the Author)

I have looked at the responses and the comments from the authors. The data is nice and the authors successfully show that QHF ferromagnetism in ABA trilayer graphene and have now also discussed valley ferromagnetism. The abstract is vastly improved and so is the introduction. Further experiments on the dual gated samples are helpful in emphasizing their zero field approximations.

The paper is nice but I still believe it lacks scientific novelty for maybe Nat. or Nat. Phys, but I think there is sufficient interesting physics here to warrant publication in Nat. Comm.

Reviewer #2 (Remarks to the Author)

In this new version of the paper called "Strong electronic interaction and multiple quantum Hall ferromagnetic phases in trilayer graphene" Professor Deshmukh and co-workers have convincingly answered to all comments I had on the previous version of the manuscript. With the new transport measurements made on a dual gated sample, I think they correctly confirmed the validity of the model they have used to describe the experimental data. I think in his present form the paper is pertinent and should have an impact on the graphene community and 2D material in general. I therefore would suggest the paper for publication in Nature Communications.

Reviewer #3 (Remarks to the Author)

Dear Authors,

Thank you for your responses to my questions. Your comprehensive and detailed answers showed that the effects reported in the manuscript are indeed electron interaction induced and are attributed to QHF.

I spent the most time debating reviewer #1's question and your response regarding the novelty of the presented results. However, the QHF has never been reported in TLG, nor has its presence in TLG ever been proven. I am now convinced that this work deserves to be published in Nature Communications.

Thank you again and I look forward to new exciting results from you.

REVIEWERS' COMMENTS:

Reviewer #1 (Remarks to the Author):

I have looked at the responses and the comments from the authors. The data is nice and the authors successfully show that QHF ferromagnetism in ABA trilayer graphene and have now also discussed valley ferromagnetism. The abstract is vastly improved and so is the introduction. Further experiments on the dual gated samples are helpful in emphasizing their zero field approximations.

The paper is nice but I still believe it lacks scientific novelty for maybe Nat. or Nat. Phys, but I think there is sufficient interesting physics here to warrant publication in Nat. Comm.

Reviewer #2 (Remarks to the Author):

In this new version of the paper called "Strong electronic interaction and multiple quantum Hall ferromagnetic phases in trilayer graphene" Professor Deshmukh and co-workers have convincingly answered to all comments I had on the previous version of the manuscript. With the new transport measurements made on a dual gated sample, I think they correctly confirmed the validity of the model they have used to describe the experimental data. I think in his present form the paper is pertinent and should have an impact on the graphene community and 2D material in general. I therefore would suggest the paper for publication in Nature Communications.

Reviewer #3 (Remarks to the Author):

Dear Authors,

Thank you for your responses to my questions. Your comprehensive and detailed answers showed that the effects reported in the manuscript are indeed electron interaction induced and are attributed to QHF.

I spent the most time debating reviewer #1's question and your response regarding the novelty of the presented results. However, the QHF has never been reported in TLG, nor has its presence in TLG ever been proven. I am now convinced that this work deserves to be published in Nature Communications.

Thank you again and I look forward to new exciting results from you.

Addressing reviewers' comments

We thank the reviewers' for reviewing our article and suggesting for publication without requiring any more changes.